# Does Folic Acid Protect Patients with Inflammatory Bowel Disease from Complications?

**DOI:** 10.3390/nu13114036

**Published:** 2021-11-12

**Authors:** Alicja Ewa Ratajczak, Aleksandra Szymczak-Tomczak, Anna Maria Rychter, Agnieszka Zawada, Agnieszka Dobrowolska, Iwona Krela-Kaźmierczak

**Affiliations:** Department of Gastroenterology, Dietetics and Internal Diseases, Poznan University of Medical Sciences, 61-701 Poznan, Poland; aleksandra.szymczak@o2.pl (A.S.-T.); a.m.rychter@gmail.com (A.M.R.); a.zawada@ump.edu.pl (A.Z.); agdob@ump.edu.pl (A.D.)

**Keywords:** folic acid, homocysteine, inflammatory bowel disease, microbiota

## Abstract

Folic acid, referred to as vitamin B9, is a water-soluble substance, which participates in the synthesis of nucleic acids, amino acids, and proteins. Similarly to B12 and B6, vitamin B9 is involved in the metabolism of homocysteine, which is associated with the *MTHFR* gene. The human body is not able to synthesize folic acid; thus, it must be supplemented with diet. The most common consequence of folic acid deficiency is anemia; however, some studies have also demonstrated the correlation between low bone mineral density, hyperhomocysteinemia, and folic acid deficiency. Patients with inflammatory bowel disease (IBD) frequently suffer from malabsorption and avoid certain products, such as fresh fruits and vegetables, which constitute the main sources of vitamin B9. Additionally, the use of sulfasalazine by patients may result in folic acid deficiency. Therefore, IBD patients present a higher risk of folic acid deficiency and require particular supervision with regard to anemia and osteoporosis prevention, which are common consequences of IBD.

## 1. Introduction

Folic acid (FA), also known as vitamin B9, is a fully oxidated synthetic form of pteroylglutamic acid monoglutamate and a water-soluble vitamin, whose name originates from the Latin “folium” meaning “leaf.” Naturally occurring folic acid has a reduced form, referred to as folate [1,2]. Since the human body is unable to produce vitamin B9 by itself, it must either be derived from a traditional or fortified diet, as the human intestinal microbiome is capable of synthesizing it [3], or by means of potential supplementation. This vitamin is essential for the growth of new cells and remethylation of homocysteine (Hcy), which is vital for the process of nucleotide synthesis [4], while appropriate vitamin B9 intake during pregnancy is a preventive factor of neural tube defects (NTDs) in gastrulation [5]. Conversely, folic acid deficiency is associated with several types of adverse health conditions, and although the most discussed conditions comprise anemia and cardiovascular disease, it is common among patients suffering from inflammatory bowel disease (IBD) [6]. Some studies have also reported an increased risk of osteoporosis as a consequence of low folic acid concentration, suggesting low bone mineral density may be due to low folic acid levels or hyperhomocysteinemia. Additionally, genetic factors are also known to affect the FA concentration [7].

## 2. The Role of Folic Acid in the Human Body

Folate performs many functions in the human body—it acts as a coenzyme in the synthesis of purines and pyrimidines, but it is also involved both in the transformation of one-carbon units and in the methylation cycle. Folate deficiency may be the result of insufficient intake, higher demand, malabsorption, or administration of certain drugs. Hence, it may lead to an increased risk of numerous diseases, such as cardiovascular disease, neoplasms, and cognitive impairment. In fact, B9 deficiency may result in hyperhomocysteinemia and disorders of protein and DNA synthesis [8]. Folic acid is absorbed by the high-affinity folate transporter in the active process in the duodenum and jejunum. In food, folates occur as polyglutamates, which may be absorbed in the intestine following enzymatic conversion into folate monoglutamates by the jejunal mucosal [9,10].

The molecular structure of folic acid comprises three different moieties—glutamic acid residue, pteroyl group, and para-aminobenzoic acid [11]. However, it must be subjected to a two-step enzymatic reaction with the dihydrofolic acid intermediate and the dihydrofolate reductase (DHFR) enzyme before it becomes an active coenzyme, i.e., the THF (tetrahydrofolian) form [5]. Subsequently, THF converts to 5-10-MTHF (methylenetetrahydrofolate), and it is reduced by MTHFR to 5-MTHF, which is involved in homocysteine conversion to methionine, donating the remaining methyl group in the process via methionine synthase. Figure 1 shows a diagram of folic acid metabolic pathways. Apart from the abovementioned elements, vitamin B12 and B6 also play a major role in the metabolism of homocysteine [12]. In fact, folic acid takes part in one-carbon metabolism, because 1-carbon units are transferred to THF for reduction or oxidation and are essential for DNA synthesis. S-adenosylmethionine is a methyl donor for biological methylation, including DNA and protein methylation. Low concentrations of folic acid in cell division inhibit the conversion of dUMP (deoxyuridine monophosphate) to dTMP (thymidylate) and uracil may be substituted in the DNA sequence [13]. Therefore, folic acid deficiency may lead to genomic hypomethylation [14]. It has been proven that cytosine methylation in the DNA sequence plays an important role in gene expression [15]. The control of gene transcription and particularly its suppression can stem from changes in methylation levels within the promoter regions of genes. Hypomethylation caused by folic acid deficiency can lead to the induction of protooncogenes, which promote tumorigenesis [16]. Hypermethylation, in turn, causes inactivation of the promoter regions of suppressor genes.

Folic acid is fundamental in the proper development of pregnancy, as its deficiency exacerbates the risk of neural tube defects in children, whereas its supplementation in the course of pregnancy mitigates the risk of heart disorders [1]. In fact, an elevated demand for vitamin B9 during gestation may lead to a decrease in its concentration. Crucially, folic acid participates in the synthesis of protein and amino acids, as well as in the multiplication of cells, which is particularly important in the first weeks of gestation. Current research indicates that the supplementation of multivitamin formulations, including folic acid, may lower the risk of pre-eclampsia in pregnant women [17]. Finally, vitamin B9, as well as B12, participate in erythropoiesis; thus, the deficiency of the aforementioned substances may lead to macrocytosis, erythroblasts apoptosis, and anemia [18].

As a result, folic acid affects numerous human metabolic pathways; therefore, inadequate folic acid intake may cause many diseases, including cancers, cardiovascular diseases, cognitive disorders, birth defects, and anemia [19].

### 2.1. Dietary Sources of Folic Acid

Green leafy vegetables, nuts, beans, and vitamin B9-supplemented products (e.g., rice, breakfast cereals, and pasta) are rich sources of folic acid [20]; Table 1 shows the folate content of the selected products [21]. It should be noted that the bioavailability of folate in food (estimated to equal approximately 50%) is approximately half of that of synthetic folic acid found in supplements [1] (amounting to 85–100%), although it is worth bearing in mind that taking supplements with meals reduces bioavailability [22]. Nevertheless, it is difficult to determine the bioavailability of folic acid in different groups of products. In fact, the supplementation of folic acid (400 µg/day) for five weeks significantly increases the serum folate levels in pregnant women [23]. An enhancement of serum folate levels upon folic acid supplementation in adults has also been observed [24].

### 2.2. Recommendations Regarding Folic Acid Intake

The daily requirements for folic acid depend on the patient’s medical condition. Folic acid, similarly to other water-soluble vitamins, does not accumulate in the human body and is rarely known to cause toxic effects [1,2]. Since the human body is unable to synthesize folate, supplementation or a dietary intake is essential. It is noteworthy that the daily intake of folic acid in food amounts to 150–250 μg, which is significantly below the recommended dietary allowance (RDA). Since folic acid plays a role in embryonic and fetal development, pregnant women, or women attempting pregnancy, its supplementation should be initiated 12 weeks prior to pregnancy and should be continued throughout the entire pregnancy, as well as during the post-partum period and breastfeeding. The recommended supplementation dosage of folic acid depends on the risk of neural birth tube defects, i.e., 0.4 mg/day for women at low risk and 5 mg/day for women in a high risk group [25]. Additionally, folic acid supplements are recommended for smokers, individuals treated with aspirin, patients suffering from kidney disease, in whom serum homocysteine levels often increase albuminuria, individuals taking certain medications following bariatric surgery, as well as in patients with diseases related to malabsorption (e.g., inflammatory bowel disease) [20].

## 3. The Role of Folic Acid in Inflammatory Bowel Disease

Folic acid deficiency constitutes a serious health factor, particularly for patients suffering from IBD. Yun et al. demonstrated in their meta-analysis that the level of folate in IBD patients was significantly lower when compared to that of healthy groups. Additionally, the folate concentration was lower in UC patients than in healthy individuals, although not in CD patients [6]. On the contrary, according to Ehrlich et al., folic acid deficiency occurred in approximately 92% of CD patients, as well as in more than 94% of UC patients and patients with an unclassified form of IBD. Moreover, 10–13% and 3.8–9.7% of children suffering from CD and UC, respectively, displayed vitamin B9 deficiency [26].

One of the significant risk factors in IBD is a poor folic acid diet, as patients often avoid vitamin B9-rich products, mostly for fear of the exacerbation of symptoms following consumption. Another one is total parenteral nutrition, as folic acid is absorbed in the duodenum and proximal jejunum. Therefore, severe inflammation, fibrosis, resection, and the occurrence of fistulas in this region may impair absorption due to a reduced active absorption area. Finally, active inflammation leads to a higher demand for folic acid, which is associated with the increased production of granulocytes and inflammatory cells [27].

Folic acid plays an important role in IBD patients—it affects the growth and regulation of cell functions because it is involved in the production of nucleic acids, protein synthesis, and amino acid transformation. IBD treatment involves the administration of methotrexate, a chemical compound included in the antimetabolite group, which has immunomodulatory and anti-inflammatory activity. The structure of methotrexate is similar to that of folic acid, and it inhibits the activity of dihydrofolate reductase, which catalyzes the transformation of dihydrofolate to tetrahydrofolate [28]. The most common adverse reactions comprise nausea, vomiting, diarrhea, bloating, liver damage, bone marrow suppression, pneumonitis, teratogenic effects, and folic acid deficiency. Studies have shown that the supplementation of folic acid may decrease the occurrence of side effects, mainly gastrointestinal disease, inflammation of the mucosa, and myelotoxicity [29]. The causes and consequences of folic acid deficiency in IBD are presented on Figure 2.

According to the guidelines of the European Crohn’s and Colitis Organization (ECCO), folic acid supplementation of 5 mg is a recommended dose within two to three days of methotrexate administration [30], while the British Society of Gastroenterology recommends discontinuation of methotrexate in women who are planning pregnancy during therapy and a high dose (15 mg daily) of folic acid supplementation for a minimum period of six months [31]. Moreover, since sulfasalazine impairs folic acid absorption, selected patients treated with sulfasalazine should be supplemented with folic acid [32]. Additionally, pregnant women suffering from IBD treated with sulfasalazine should increase their dose of folic acid to 2 mg/day [33]. In the case of IBD, folic acid deficiency is one of the most common causal agents of non-iron deficiency anemia (NIDA) [34], which may decrease the quality of life in patients. Simultaneously, patients suffering from IBD are at a higher risk of folic deficiency (e.g., with inflammation in the small intestine or following resection) and require adequate supervision.

In folic acid deficiency, the production of red blood cells is disturbed, their volume increases, the survival time is shortened, and the bone marrow is damaged prematurely. The main reason for this phenomenon is the disruption of the synthesis of nucleic acids, mainly purine precursors. In normal circumstances, the level of folic acid should be controlled at least once a year, whereas in patients following extensive resection of the small intestine, extensive involvement of the ileum, with intestinal reservoir or with symptoms of deficiency its levels, should be monitored more frequently [35,36].

According to the guidelines of ECCO-ESGAR (the European Society of Gastrointestinal and Abdominal Radiology), the folic acid concentration should be tested every three to six months in patients suffering from IBD with a damaged small intestine, or following small intestine resection [37]. Samblas et al. suggested that folic acid may be effective in controlling the chronic inflammation in inflammatory diseases, mainly by means of DNA methylation. Folic acid is a methyl donor, and this activity is associated with the synthesis of S-adenosyl methionine (SAM), which is also a methyl donor. As researchers have observed, folic acid and other methyl donors decreases the expression of interleukin 1β (IL-1β), as well as tumor necrosis factor (TNF). Moreover, they also reduce the level of C-C motif chemokine ligand 2 (*CCL2*) mRNA, and increases the methylation in CpGs located in the genes of IL-1β, SERPINE1, and IL-18 [38].

Patients with long-term UC or CD present an increased risk of colorectal cancer (CRC) associated with inflammation and dysplasia. Additionally, CRC related to IBD is associated with chromosomal instability, microsatellite instability, and hypermethylation. In sporadic CRC, a low intake of folic acid is associated with a higher risk of adenomas and CRC, and the mechanisms are possibly related to the maintenance of normal methylation of DNA [39]. DNA methylation may affect gene expression. Therefore, folic acid deficiency might cause hypomethylation of DNA, leading to disorders of protooncogene participating in cancerogenesis. According to another hypothesis, deficiency of folic acid may induce uracil misincorporation during the synthesis of DNA, which leads to the breakage of DNA strands and chromosome damage [16].

According to Lashner et al., the supplementation of folic acid decreases the risk of neoplasms by approximately 62%, although these changes are not significant. Moreover, researchers have reported no considerable correlation between the folic acid dose and the development of neoplasms in a six-month supplementation of folate [40,41]. In contrast, as pointed out in the experimental study by Biasco et al., a three-month long supplementation of folic acid resulted in decreased cell proliferation [39]. A meta-analysis indicated that the supplementation of folic acid plays a protective role against the development of colorectal cancer [42]. It is vital to note that the role of folic acid in the prevention of CRC in patients suffering from IBD has not been substantiated in a large, randomized study yet. Furthermore, ECCO does not recommend routine supplementation of folic acid for the prevention of CRC in patients suffering from IBD [43].

Hyperhomocysteinemia, associated with group B vitamin deficiency, may increase the activity of Th17 cells in the bowel mucosa [44]. It is suggested that the supplementation of folic acid—due to decreased homocysteine levels—may reduce the activity of IBD and the occurrence of other autoimmune diseases [45]. The supplementation of folic acid, vitamin B12, and vitamin D did not change the lumbar spine and femoral neck BMD values more than in the placebo group (supplemented with vitamin D only). However, the concentration of Hcy was significantly reduced in the study group when compared to patients administered the placebo [46]. As Salari et al. reported, a six-month folic acid supplementation in postmenopausal women resulted in a significantly lower vitamin B12 level and a higher concentration of osteocalcin in the study group as compared to the control group [47].

### 3.1. Methylenotetrahydrofolate Reductase Gene

The *MTHFR* gene is located in the shorter arm of chromosome 1 (loci: 1p36.22) [48]. The metabolism of Hcy depends on MTHFR, which is coded by the *MTHFR* gene. A single nucleotide polymorphism (SNP) in the position 677 C → T causes a decrease in enzyme activity by 60% [49]. As pointed by Hanks et al., the 677CC genotype occurred in 55% of subjects, 677CT in 35%, and 677TT in 10% [50]. It is interesting to observe that the 677TT genotype occurred in 17.5% of patients presenting with ulcerative colitis (UC), in 16.8% of subjects with Crohn’s disease (CD), and in 7.3% of healthy individuals. Moreover, in patients suffering from inflammatory bowel disease (IBD), the Hcy levels were significantly higher in persons with the 677TT genotype when compared to those with 677CT [51]. Nevertheless, no significant differences were observed in the frequency of the 677TT genotype in IBD patients and healthy persons [52,53]. As a meta-analysis showed, the mutation of the *MTHFR* gene in the position 677C/T did not increase the risk of IBD development [54], although in the Chinese population, 677TT polymorphism occurs more frequently in pancolitis than in other CD patients [55]. The mutation has been associated with a higher serum level of homocysteine; in fact, subjects with the 677TT and 677CT genotypes present higher homocysteine levels than subjects with 677CC. Moreover, serum folate levels decreased as the number of T alleles increased [56]. Therefore, using 677CC and 677CT as reference values, the odds ratio for folic acid deficiency amounted to 2.34 for individuals with the 677TT genotype. [57].

Another polymorphism of the *MTHFR* gene is 1298A → C. It is vital to notice that individuals with 1298AA and 1298AC demonstrate higher folic acid concentrations than patients with 1298CC, which is considered a wild-type [57].

Additionally, the differences in the previously discussed polymorphisms vary in different populations; for instance, the frequency of allele 1298C and genotype 1298CC is lower in western Africa and Mexico than in European countries (Italy—Sicily and France). Moreover, the 677TT genotype considerably increases the Hcy levels in individuals from western Africa, whereas only moderately in subjects from France and Italy. However, the abovementioned genotype does not elevate the concentration of Hcy in Mexicans [58].

Patients suffering from IBD are thought to be at a risk of developing osteoporosis, although no research has been conducted in regard to the risk of low bone mineral density (BMD) in IBD patients and the specific polymorphisms of the *MTHFR* gene. Gjesdal et al. presented various genotypes of the MTHFR gene in position 677C/T, and claimed that 1298A/C did not affect the risk of hip fracture [59]. On the contrary, as another meta-analysis showed, postmenopausal women with 677TT presented a higher femoral neck BMD than women with 677CC/CT, in spite of the fact that no such observation was made concerning lumbar spine BMD [60]. Furthermore, the TT genotype occurred more frequently in women with vertebral fractures than in female patients without such fractures [61]. Finally, studies have shown that the 677C/T polymorphism also affects spine BMD in nine-year-old children, which may further affect peak bone mass [62].

### 3.2. Homocysteine and Bone Mineral Density in Inflammatory Bowel Disease Patients

Inflammatory bowel disease patients present an increased risk of low BMD, leading to osteoporosis. Among others, risk factors of osteoporosis in IBD include malnutrition, low body mass, malabsorption, or use of corticosteroids [63]. According to Adriani et al., osteopenia and osteoporosis affect 46% and 11% of patients suffering from IBD, respectively [64].

Research has indicated that the mean homocysteine concentrations are, in general, higher in IBD patients than in control groups, with no significant difference between UC and CD; however, it has been pointed out that Hcy levels are higher in men than in women and also correlate with age in the study group, although not in the control group [65,66]. Similarly, Akbult et al. noted that the Hcy level among patients suffering from UC was higher than in healthy individuals [67], while Zezes et al. demonstrated a greater incidence of hyperhomocysteinemia in UC patients than in healthy subjects [68].

Homocysteine (Hcy) is an amino acid containing sulfur and does not occur in proteins, but it is formed in the course of methionine transmethylation [69], and the average Hcy level is below 15 μmol/L. Vitamin B6, folic acid, vitamin B12, and methylene tetrahydrofolate reductase are essential for homocysteine remethylation. If a deficiency of the abovementioned substances occurs, homocysteine is accumulated in serum, leading to hyperhomocysteinemia; moreover, it may be metabolized into cysteine and excreted in urine [70]. Homocysteine is a well-known risk factor of atherosclerosis, which is a chronic inflammation of the endothelium with an increased plasma permeability and lipid deposition in the atherosclerotic plaque. The accumulated atherosclerotic plaque, in turn, undergoes calcification and fibrosis. Additionally, homocysteine affects bone tissue, since the substance increases the activity of osteoclasts and inhibits apoptosis. The increased activity of osteoclasts may lead to increased bone resorption, a reduction of BMD, and an elevated risk of fracture [69]. What is more, Hcy exacerbates apoptosis in bone marrow and osteoclastogenesis, simultaneously decreasing blood flow in bone tissue, which may lead to reduced BMD [71], as well as increases intracellular reactive oxygen spices, which causes an increased differentiation of osteoclasts [72]. Finally, the concentration of Hcy is associated with bone resorption markers, e.g., β-CTX (*C*-terminal cross-linked telopeptide of type I collagen) [73].

No association between folic acid and vitamin B12 treatment to reduce the Hcy level and risk of hip fracture was shown in a previous meta-analysis [74]. On the contrary, Hcy levels have been shown to be correlated negatively with BMD among women [75]. In a study by Bailey et al., the researchers observed a lower serum level of vitamin B12 and folic acid in erythrocytes, as well as a higher level of bone turnover markers (serum of alkaline phosphatase and urine excreted of *N*-terminal cross-linked telopeptide of type I collagen) in postmenopausal women with a higher concentration of Hcy in comparison to individuals presenting normal Hcy levels. Additionally, Hcy levels correlated negatively with total BMD and BMD of the lumbar spine [76]. Nevertheless, Tariq et al. reported no association between Hcy levels and T- and Z-scores in postmenopausal women [77], and, according to Mittal et al., the concentration of Hcy is correlated positively with the level of parathormone (PTH) and phosphate. In contrast, Hcy levels were not related to BMD of the hip, lumbar spine, and forearm [78]. Postmenopausal women with osteoporosis presented a higher level of homocysteine than the control group. Moreover, the concentration of folic acid was lower in the study group than in the controls, although not significantly. In fact, Hcy levels were associated with the concentration of PTH, CTX (type I collagen C-telopeptides), and bone-specific alkaline phosphatase [79,80]. The adjusted hazard ratio for fracture was 2.42 (women) and 1.37 (men) for a concentration of homocysteine above 15 μM when compared to Hcy levels below 9 μM. The Hcy level correlated positively with the risk of fracture and, in fact, there was no association between the MTHFR genotype, vitamin B level, and risk of fracture [59]. As a meta-analysis showed, an increase in the homocysteine level by 1 μM elevates the risk of fracture by 4% [80]. According to Stone et al., the supplementation of folic acid and vitamins B6 and B12 did not alter the risk of fracture in women with higher levels of Hcy, or lower concentrations of folic acid, vitamin B6, or vitamin B12. Supplementation did not affect bone turnover markers such as CTX and P1NP (type I procollagen N-propeptide) [81]. Additionally, the supplementation of folic acid and vitamins B12 and B6 (individually or in combination) did not change the concentration of bone turnover markers and osteoporotic fracture in individuals suffering from the vascular disease with a normal level of Hcy [82].

Although the studies describing the association between IBD and homocysteine focused on non-IBD individuals, they suggest that hyperhomocysteinemia may constitute a potential additional factor of low BMD among IBD patients.

## 4. Microbiota and Folate Metabolism in IBD Patients

There are many factors that can affect the gut microbiota in patients with inflammatory bowel disease [83], such as age, inhabitable environment, culture habits, medical history, and the applied treatment. Nevertheless, the most critical factor affecting the diversity of species in the gut microbiota is diet, where saturated fatty acids and sugar have demonstrated a negative impact [84]. On the contrary, a vegetarian diet, rich in fiber, fruits, and vegetables, stimulates the growth of eubiotic bacteria and improves the function of enterocytes. The type of diet is particularly important for patients suffering from IBD, as a well-balanced one provides all macro- and micronutrients and eliminates products exacerbating symptoms of the disease. Conversely, fiber-rich products, vegetables, and fruits, which influence the microbiota content and the metabolism of folate, often exacerbate the symptoms. Additionally, using bacteriostatic and immunosuppressive drugs inhibiting the synthesis of PGE2 and leukotrienes may modify the gut microbiota and affect species diversity. In contrast, dietary carbon, nitrogen, water, and other nutrients provide a healthy development of intestinal bacteria, which are a source of vitamins and macro- and micronutrients. The most significant amount of folic acid is found in plant products, but folate may also be provided through animal products and various supplements [85], though the absorption of folic acid from plant products is decreased due to the presence of a conjugate inhibitor. Moreover, although the bioavailability of folic acid is higher in animal products, the content of vitamin B9 is significantly lower. As a result, the gut microbiota constitutes an essential element of folate absorption, which is twofold and occurs via folate receptors, as well as by means of specific receptors [86].

Most bacterial strains of *Bifidobacteria*, except for *B. gallicum* and *biavatii*, possess genes responsible for folate synthesis [87], while most lactic acid bacteria cannot synthesize folate. In fact, *Lactobacillus plantarum* is the only strain capable of producing folate, although the presence of 4-aminobenzoic acid is necessary [86]. According to Strozzi et al., the supplementation of *Bifidobacterium adolscentis* and *pseudocatenulatum* results in an increased level of fecal folate [88]. Additionally, as pointed out by LeBlanc et al., lactic acid bacteria and *Bifidobacteria*—found in fermentable dairy products—are able to synthesize vitamins de novo, particularly group B vitamins such as folic acid and vitamin B12 [89]. Furthermore, the supplementation of *L. plantarum* decreases the level of Il-8 and TNF-α and the expression of the occludin gene [90]. MacFarlane, on the other hand, evaluated how diet and microflora affect the course of IBD. The effect of folate on colonic microflora and the development of colonic tumors was determined in chemically induced ulcerative colitis in mice. According to the study results, the concentration of folic acid depended on the diet, and mice with colitis did present lower levels of circulating folic acid. However, folic acid had a minimal effect on tumor initiation and no effect on intestinal microflora. These data suggest that folic acid intake has little or no effect on the alleviation of IBD symptoms, or the risk of developing colon cancer in patients with IBD [91].

In another study, folate synthesized in the colon was absorbed and utilized by the host, and the local production of folic acid in the colon could help patients with IBD and reduce the risk of carcinogenesis [92]. In a study by Laiño, *Lactobacillus delbrueckii* subsp. *bulgaricus* CRL 863 and *S. thermophilus* CRL 415 and CRL 803 produced folic acid in fat-free milk, and increased the initial concentration of folic acid by approximately 190% [93]. Therefore, probiotic supplementation containing *Bifidobacterium* and *Lactobacillus plantarum* strains may increase folic acid production in patients with IBD and have a protective function for colonocytes in the course of this disease. Additionally, *Bifidobacteria* are involved in the regulation of intestinal homeostasis and have the ability to modulate the immune response [94]. Finally, it has been demonstrated that the use of *B. adolescentis* and *B. pseudocatenulatum* strains in humans increases the fecal folic acid concentration [95].

Microbial folate synthesis may also be affected by certain drugs, e.g., metformin, used in type 2 diabetes mellitus and insulin resistance, decreasing folate synthesis due to an increase of *Coenorhabditis elegans*, which also leads to decreasing serum folate levels [96]. Similarly, sulfonamides (structural analogs of p-aminobenzoic acid (PABA)) inhibit the synthesis of dihydrofolate (DHF) [97].

In summary, a proper supply of folic acid has a beneficial impact on the development of the gut microbiota, and certain bacteria strains provide an optimal level of folate in patients suffering from IBD.

## 5. Summary and Conclusions

Folic acid is a water-soluble group B vitamin, and its deficiency may lead to clinical complications, especially among patients suffering from IBD. Folic acid participates in the metabolism of homocysteine, high levels of which are associated with an increased risk of cardiovascular diseases and osteoporosis. Additionally, vitamin B9 is essential for the synthesis of nucleic acids and proteins. Therefore, providing adequate amounts of folic acid may prevent complications in this specific group of patients. Nevertheless, research regarding the association between folic acid, IBD, and bone mineral density is scarce. Therefore, future studies are necessary in order to investigate the potential benefits of folic acid use among IBD patients, which may improve both the course of the disease and the quality of life.

Summary:In patients suffering from IBD, the concentration of folic acid should be evaluated more frequently than once per year, as it will help to diagnose a potential deficiency and macrocytic anemia.Following the recommendations of ECCO, IBD patients treated with methotrexate should be supplied with 5 mg of folic acid at two- to three-day intervals during the administration of methotrexate.Pregnant women, or women attempting pregnancy, should supplement folic acid. The recommended dosage is 0.4–5 mg/day (depending on the risk of neural birth tube defects).The supplementation of folic acid may be a protective factor against the development of CRC. However, this hypothesis requires further research.

## Figures and Tables

**Figure 1 nutrients-13-04036-f001:**
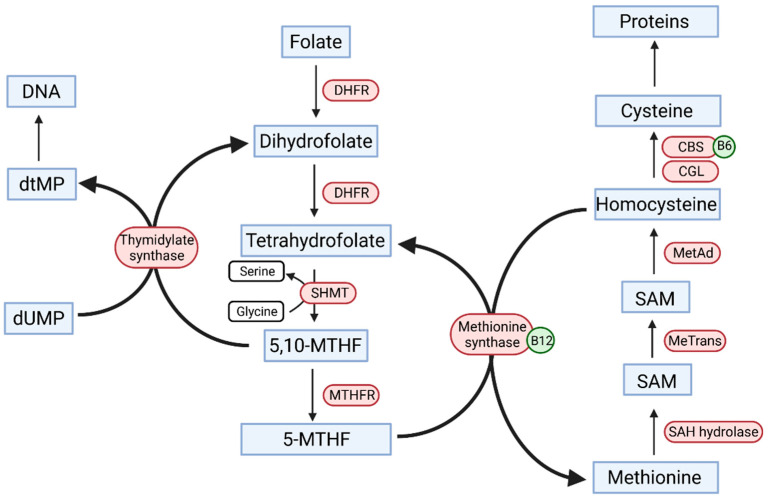
Folic acid cycle and homocysteine metabolism. DNA, deoxyribonucleic acid; dTMP, deoxythymidine monophosphate; dUMP, deoxyuridine monophosphate; DHFR, dihydrofolate reductase; SHMT, serine hydroxymethyltransferase; 5,10-MTHF, 5,10-methylenetetrahydrofolate; 5-MTHF, 5-methylenetetrahydrofolate; CBS, cystathionine-β-synthase; CGL, cystathionine gamma-lyase; MetAd, methionine adenosyltransferase; MeTrans, methyltransferase; SAM, S-adenosylomethionine; SAH, S-adenosylhomocysteine hydrolase.

**Figure 2 nutrients-13-04036-f002:**
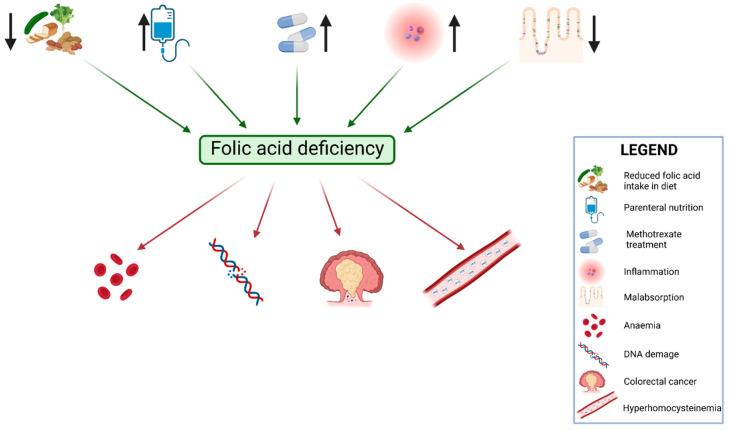
Causes and consequences of folic acid deficiency.

**Table 1 nutrients-13-04036-t001:** Folate content in the selected products [21].

Product	Folate Content in 100 g of a Product (μg)
Milk	5
Quark	27
Egg yolk	152
Chicken liver	590
Beef liver	330
Rice	29
Broccoli	119
Parsley	170
Spinach	193
Avocado	62
Apple	6

## Data Availability

Not applicable.

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
