# Peer review of "Does Folic Acid Protect Patients with Inflammatory Bowel Disease from Complications?"

_nutrients, 2021, doi:10.3390/nu13114036_

Round 1

Reviewer 1 Report

The major problem with this review is that it contains an abundant body of information on folic acid metabolism and basic and clinical science related studies, but a relevant part of the paper doesn't relate this information. direceteley to the context of IBD or is too detailed and therefore de-focuses from the IBD perspective. The manuscript would clearly benefit from deleting a substantial part of this not-IBD related information and focus on data that has a proven/direct relation to IBD.  On the other side, key statements are not supported by adequate references. This would also redistribute the "reference burden" of 100 references to the really related ones.

Below just a few more detailed remarks:

Chapter 3:

Condensate the extensive discussion on homocysteine (chapter 3.2.) with a clear focus on IBD (not on Alzheimer, not-IBD-related osteoporosis, not- IBD-related cardiovascular disease, etc. – or relate it to IBD).

Reference 29 does not support all statements in the paragraph (l168 ff). Please provide the corresponding sources.

There are no references indicated concering influence of homocysteinemia on immunological chances, e.g. Th17 cell levesla, influence on IBD and other autoimmune diseases (l.207 ff).

Chapter 4

Almost no references in the first 20 lines of the microbiota chapter (Chapter 4) . On the other side lots of information that doesn´t focus directly on IBD and folic acid nor helps really in the argumentation (e.g. l367 to 381, 397-401 or ref. 98, etc.).

In addition some other minors:

Table 1 is missing (see reference in l 96)

Repetitive/redundant sentences ( l 97)

l 127 What is the control group?

Author Response

Poznań, 02.11.2021

Alicja Ewa Ratajczak
Department of Gastroenterology, Dietetics and Internal Diseases
Poznan University of Medical Science

Dear Reviewer,

On behalf of the authors' manuscript “Does folic acid protect patients with inflammatory bowel disease from complications?”, I appreciate your helpful comments.

We feel that the manuscript is now greatly improved. In accordance with the guidelines, we have introduced the following changes:

The major problem with this review is that it contains an abundant body of information on folic acid metabolism and basic and clinical science related studies, but a relevant part of the paper doesn't relate this information. direceteley to the context of IBD or is too detailed and therefore de-focuses from the IBD perspective. The manuscript would clearly benefit from deleting a substantial part of this not-IBD related information and focus on data that has a proven/direct relation to IBD.  On the other side, key statements are not supported by adequate references. This would also redistribute the "reference burden" of 100 references to the really related ones.

Thank you very much for this comment. We changed our manuscript point by point according your suggestions and we hope that our manuscript is really well improved. 

Condensate the extensive discussion on homocysteine (chapter 3.2.) with a clear focus on IBD (not on Alzheimer, not-IBD-related osteoporosis, not- IBD-related cardiovascular disease, etc. – or relate it to IBD).

Thank you very much for this comment. We modified this section and delated information about non-osteoporosis and non-IBD disease. Additionally, there are few study referring to association between folic acid and IBD-related osteoporosis. Therefore, we think that discuss about non-IBD-related osteoporosis and folate may be interesting point for discussion and future studies. Moreover, we hope this section is more clearly now.

Reference 29 does not support all statements in the paragraph (l168 ff). Please provide the corresponding sources.

Thank you very much for this comment. We added table 1.

There are no references indicated concering influence of homocysteinemia on immunological chances, e.g. Th17 cell levesla, influence on IBD and other autoimmune diseases (l.207 ff).

The indicated references has been added, which we marked in manuscript.

Almost no references in the first 20 lines of the microbiota chapter (Chapter 4) . On the other side lots of information that doesn´t focus directly on IBD and folic acid nor helps really in the argumentation (e.g. l367 to 381, 397-401 or ref. 98, etc.).

Thank you very much for this comment. We rebuilt this section and we hope its more clearly.

l 127 What is the control group?

Thank you very much for this comment. We changed “control” on “healthy”.

Additionally, in the preparation of this manuscript we have cooperated with a biomedical translation company – TranslationLab, with whom we collaborated in the course of the publication of another paper (in Journal of Clinical Medicine or Nutrition). All changes have been marked in the manuscript.

We would like to express our thanks for the cooperation, and we hope that our paper after corrections can undergo a further review process and be successfully published in Nutrients.

Yours faithfully,

Alicja Ratajczak

Reviewer 2 Report

Study by Alicja Ewa Ratajczak et al- Entitled “Does folic acid protect patients with inflammatory bowel disease from complications?” is elaborate and informative. Although, there are limited studies which are showing some relation between folic acid and IBD. Also, there are some concerns that need to be addressed to strengthen the article.

 Concerns:

  1. It would be good if the author can mention any limitations and future directions in this article.
  2. The theory behind DNA methylation and the role of folic acid in methylation of genes should be explained as it requires more detail.
  3. It would be good if the author can also incorporate more visuals/diagrams into the article to make it easier to understand to the reader.
  4. Lines 40-42 should be reworded for better flow of information.
  5. Line-50 requires a citation.
  6. Line-110, author should make a background as to why the folic acid supplementation is important in kidney diseases.
  7. Line-136, the author should explain the absorption process of folic acid. For example, does it require any receptor or transporter.
  8. Line-217 and line-257, 3.1 and 3.2 have the same title. Please clarify.
  9. Line-342 through line-344, and line-391 are confusing to read. Please rephrase the sentences for better readability.
  10. Authors should add an abbreviation table.
  11. In summary and conclusion part, point 4, the authors make the statement that folic acid may be a protective factor against CRC development. However, as authors have mentioned above, the folic acid is a methyl donor and induces methylation which is a contradictory finding in CRC.

Author Response

Poznań, 02.11.2021

Alicja Ewa Ratajczak
Department of Gastroenterology, Dietetics and Internal Diseases
Poznan University of Medical Science

Dear Reviewer,

On behalf of the authors' manuscript “Does folic acid protect patients with inflammatory bowel disease from complications?”, I appreciate your helpful comments.

We feel that the manuscript is now greatly improved. In accordance with the guidelines, we have introduced the following changes:

It would be good if the author can mention any limitations and future directions in this article.

Thank you very much for this comment. We added suggested information in section “Summary and conclusions)

The theory behind DNA methylation and the role of folic acid in methylation of genes should be explained as it requires more detail.
In summary and conclusion part, point 4, the authors make the statement that folic acid may be a protective factor against CRC development. However, as authors have mentioned above, the folic acid is a methyl donor and induces methylation which is a contradictory finding in CRC.

Thank you very much for this comment. We added more precise information about role of folic acid in methylation.

It would be good if the author can also incorporate more visuals/diagrams into the article to make it easier to understand to the reader.

Thank you very much for this comment. We added one more diagram.  

Line-50 requires a citation

The indicated reference has been added, which we marked in manuscript.

Line-110, author should make a background as to why the folic acid supplementation is important in kidney diseases.

Thank you very much for this comment. We added suggested information.

Line-136, the author should explain the absorption process of folic acid. For example, does it require any receptor or transporter.

Thank you very much for this comment. We described process of folic acid absorption in section “The role of folic acid in the human body”.  

Line-217 and line-257, 3.1 and 3.2 have the same title. Please clarify.

Thank you very much for this comment. We changed title of 3.2. Now, the title of this section is “Homocysteine and bone mineral density in inflammatory bowel disease patients”.

Authors should add an abbreviation table.

Thank you very much for this comment. However, according to guidelines for authors, table of abbreviation is not recommended.

Additionally, in the preparation of this manuscript we have cooperated with a biomedical translation company – TranslationLab, with whom we collaborated in the course of the publication of another paper (in Journal of Clinical Medicine or Nutrition). All changes have been marked in the manuscript.

We would like to express our thanks for the cooperation, and we hope that our paper after corrections can undergo a further review process and be successfully published in Nutrients.

Yours faithfully,

Alicja Ratajczak
